# GZ17-6.02 Inhibits the Growth of EGFRvIII+ Glioblastoma

**DOI:** 10.3390/ijms23084174

**Published:** 2022-04-10

**Authors:** Justin Choi, Zachary A. Bordeaux, Jaimie McKeel, Cory Nanni, Nishadh Sutaria, Gabriella Braun, Cole Davis, Meghan N. Miller, Martin P. Alphonse, Shawn G. Kwatra, Cameron E. West, Madan M. Kwatra

**Affiliations:** 1Department of Dermatology, Johns Hopkins University School of Medicine, Baltimore, MD 21205, USA; jchoi222@uic.edu (J.C.); zbordeaux1@pride.hofstra.edu (Z.A.B.); nsutari1@jhmi.edu (N.S.); malphon1@jhmi.edu (M.P.A.); skwatra1@jhmi.edu (S.G.K.); 2Department of Anesthesiology, Duke University School of Medicine, Durham, NC 27710, USA; jaimie.mckeel@duke.edu (J.M.); cory.nanni@gmail.com (C.N.); gabriella.braun@duke.edu (G.B.); cole.davis@duke.edu (C.D.); meghan.n.miller@duke.edu (M.N.M.); 3Department of Oncology, Johns Hopkins University School of Medicine, Baltimore, MD 21205, USA; 4Genzada Pharmaceuticals, Hutchinson, KS 67502, USA; cameron.west@genzada.com; 5Department of Pharmacology and Cancer Biology, Duke University School of Medicine, Durham, NC 27710, USA

**Keywords:** glioblastoma, glioblastoma stem cells, EGFR, EGFRvIII, super-enhancer

## Abstract

Epidermal Growth Factor Receptor (EGFR) is amplified in over 50% of glioblastomas and promotes tumor formation and progression. However, attempts to treat glioblastoma with EGFR tyrosine kinase inhibitors have been unsuccessful thus far. The current standard of care is especially poor in patients with a constitutively active form of EGFR, EGFRvIII, which is associated with shorter survival time. This study examined the effect of GZ17-6.02, a novel anti-cancer agent undergoing phase 1 studies, on two EGFRvIII+ glioblastoma stem cells: D10-0171 and D317. In vitro analyses showed that GZ17-6.02 inhibited the growth of both D10-0171 and D317 cells with IC_50_ values of 24.84 and 28.28 µg/mL respectively. RNA sequencing and reverse phase protein array analyses revealed that GZ17-6.02 downregulates pathways primarily related to steroid synthesis and cell cycle progression. Interestingly, G17-6.02’s mechanism of action involves the downregulation of the recently identified glioblastoma super-enhancer genes *WSCD1, EVOL2*, and *KLHDC8A.* Finally, a subcutaneous xenograft model showed that GZ17-6.02 inhibits glioblastoma growth in vivo. We conclude that GZ17-6.02 is a promising combination drug effective at inhibiting the growth of a subset of glioblastomas and our data warrants further preclinical studies utilizing xenograft models to identify patients that may respond to this drug.

## 1. Introduction

Glioblastoma is a diffusely infiltrative malignant brain tumor that leads to significant morbidity and mortality among affected patients. The 5-year survival rate of glioblastoma is approximately 6.8%, and median survival ranges between 15 and 18 months [1]. The current treatment approach combines tumor resection, radiation, and chemotherapy; however, treated tumors invariably recur within 2 years [2]. Despite increased understanding of the molecular pathogenesis and genomics of glioblastoma, little progress has been made in improving outcomes [1,3,4,5].

One particularly promising therapeutic target in glioblastoma is the epidermal growth factor receptor (EGFR). Amplification of EGFR is noted in over 50% of glioblastoma cases [6], and has tumor-promoting effects on multiple downstream signaling pathways that are linked to proliferation, angiogenesis, fatty acid synthesis, infiltration, and apoptosis [7,8,9,10]. However, attempts to treat glioblastoma with targeted EGFR tyrosine kinase inhibitors (TKIs), such as erlotinib, gefitinib, lapatinib, and afatinib, have remained largely unsuccessful [10,11,12]. This is attributed to two main factors. First, recent studies indicate that EGFR in glioblastoma occurs in several forms, including EGFRvIII, EGFR with kinase domain duplication (EGFR-KDD), EGFR fused with SEPT14 (EGFR-SEPT14), and EGFR with point mutations in both intracellular and extracellular domains [6,11,12,13]. Second, most of the previously tested EGFR-TKIs did not penetrate the brain well [11,12]. Thus, targeting of EGFR in glioblastoma remains to be properly evaluated by examining novel agents that penetrate the brain well against specific molecular forms of EGFR [11].

The present study was undertaken to evaluate GZ17-6.02, a novel anti-cancer agent which is currently undergoing phase I trials in various cancers [14,15,16,17], for its growth inhibition of glioblastomas expressing a specific form of EGFR, namely EGFRvIII. This form of EGFR is present in 20–25% of glioblastoma patients [9,18,19,20,21], and in patients with EGFR amplification, those carrying the EGFRvIII mutation show a significantly shorter survival time than patients without this mutation [22]. EGFRvIII differs from the wild-type EGFR in that it cannot bind epidermal growth factor (EGF), and its kinase domain is constitutively active [9,18,19,20,21].

GZ17-6.02, a synthetic mixture of three compounds including curcumin (10% by weight), harmine (13% by weight), and isovanillin (77% by weight) [23], has shown anti-tumor activity against several cancers, including head and neck squamous cell carcinoma, pancreatic cancer, melanoma, breast cancer, and non-small cell lung cancer [14,15,16,17,23,24,25]. This agent was chosen to test its ability to inhibit the growth of EGFRvIII+ glioblastoma because of the hypothesized synergistic effect of its components in disrupting EGFR signaling. Curcumin, the main active compound in the dietary spice turmeric, is well studied as an antitumor agent that inhibits tumor growth by multiple mechanisms such as the suppression of PI3K-AKT/mTOR signaling [26,27]. Additionally, curcumin disrupts EGFR signaling by inhibiting the binding of the transcription factor, early growth response-1 (Egr-1), to the EGFR promoter, suppressing *egr-1* expression, and inhibiting the phosphorylation of EGFR and several downstream signaling molecules [28,29,30].

Harmine is the most bioactive component of the medicinal plants *Paganum harmala* and *Arum palaestinum*¸ which have been used for centuries to treat many ailments including cancer [24]. Harmine exerts potent antitumor effects through multiple mechanisms such as the induction of DNA damage, inhibition of DNA replication, inhibition of cell cycle progression, and suppression of angiogenesis [31]. In glioblastoma specifically, harmine suppresses EGFR-dependent growth through the inhibition of the dual-specificity tyrosine-(Y)-phosphorylation-regulated kinase, DYRK1A [32].

Finally, isovanillin is a product found in several plant species that has antioxidant effects and is proposed as an antitumor agent [33,34,35]. Furthermore, isovanillin is thought to potentiate the antitumor activity of curcumin and harmine by forming a tight complex with these compounds resulting in an entity with unique biologic properties compared to curcumin or harmine alone [14].

The EGFRvIII+ glioblastomas selected for this study are D10-0171 and D317, which we had used in our recent study evaluating the efficacy of osimertinib against EGFRvIII+ glioblastoma [12]. We find that GZ17-6.02 inhibits the growth of both glioblastoma stem cells (GSCs) in cell culture models and the mechanism of action appears to be the downregulation of superenhancer genes implicated in glioblastoma growth.

## 2. Results

### 2.1. GZ17-6.02 Inhibits the Growth and Invasion of EGFRvIII+ GSCs

The overall study design is shown in Figure 1. Results from CellTiter-Glo viability assays are shown in Figure 2a. As can be seen, GZ17-6.02 inhibits the growth of both D10-0171 and D317 GSCs with comparable IC_50_ values: 24.84 µg/mL for D10-0171 and 28.28 µg/mL for D317. We next evaluated the ability of GZ17-6.02 to suppress the invasive behavior of D10-0171 and D317 using a Matrigel invasion assay. As Figure 2b shows, the invasion of both D10-0171 and D317 is inhibited by GZ17-6.02. ImageJ was used to quantify the data obtained from the invasion assay. This analysis demonstrated a significant decrease in area occupied by tumor spheroids and their radial projections following GZ17-6.02 treatment in both cell lines. For D10-0171, a GZ17-6.02 concentration of 25 µg/mL resulted in a decrease in area occupied by tumor from (mean ± standard deviation) 40.75 ± 1.92% for control cells to 33.83 ± 1.94% for treated cells (*p* = 0.023), while a GZ17-6.02 concentration of 50 µg/mL resulted in a decrease in area occupied by tumor to 7.49 ± 2.63% (*p* < 0.001) (Figure 2c). For D317, a GZ17-6.02 concentration of 50 µg/mL resulted in a decrease in area occupied by tumor from 20.12 ± 1.98% for control cells to 12.94 ± 1.24% in treated cells (*p* = 0.012) (Figure 2d). This assay indicates that while GZ17-6.02 inhibits the growth and invasion of both D10-0171 and D317, its effect on D10-0171 is much more pronounced.

### 2.2. Molecular Differences between D10-0171 and D317

In our recent study evaluating the efficacy of osimertinib against EGFRvIII+ glioblastoma, we found that compared to D317, D10-0171 exhibited increased expression of growth-promoting GSC markers such as OLIG2, and resistance to growth inhibition by EGFR-TKIs [12]. Due to these observations, we theorized that D10-0171 would show greater invasive capacity than D317. In line with this hypothesis, the Matrigel invasion assay demonstrated that these GSC lines differ dramatically in their invasiveness, with D317 demonstrating minimal invasion and D10-0171 exhibiting highly invasive behavior. In this respect, the D10-0171 cell line is a better representative of glioblastoma because infiltration into normal brain tissue is the hallmark of this disease.

To understand the molecular variation leading to the observed differences in invasiveness between D10-0171 and D317, we next analyzed these cells using RNA sequencing (RNA-seq) and reverse phase protein arrays (RPPA). As shown in Figure 3a, initial exploratory data analysis of the RNA-seq data by principal component analysis (PCA) demonstrated a clear separation between D10-0171 and D317. Differential expression analysis revealed 7535 differentially expressed genes (DEG) in D10-0171 compared to D317, with 4576 showing upregulation and 2959 showing downregulation, as is shown in the volcano plot (Figure 3b) and heatmap (Figure 3c).

To identify the pathways that are upregulated in D10-0171 compared to D317, we also analyzed the transcriptomic data using gene set enrichment analysis (GSEA) with the hallmark gene set collection as reference. This analysis revealed upregulation of 2 pathways in D10-0171 compared to D317: cholesterol homeostasis and notch signaling (Figure 3d), both of which have been implicated in glioblastoma growth and progression [37,38,39,40].

To further elucidate the molecular differences between these two cell lines, we also obtained the phosphoproteomic profiles of D10-0171 and D317 using RPPA technology. Analysis of the phosphorylated proteins showing increased expression in D10-0171 compared to D317 using Metascape [36] revealed that ErbB (EGFR) and insulin signaling are upregulated in D10-0171 compared to D317 (Figure 3e). Taken together, these data indicate that D10-0171 and D317 possess different baseline molecular characteristics, even though they both express EGFRvIII.

### 2.3. Transcriptomic and Proteomic Effects of GZ17-6.02 on EGFRvIII + GSCs

To identify the molecular pathways/targets affected by the exposure of D10-0171 and D317 to GZ17-6.02, control and GZ17-6.02-treated GSCs were subjected to transcriptomic and phospho-proteomic analyses. A PCA plot of the transcriptomic data demonstrated a clear separation between the control and GZ17-6.02 treated D10-0171 and control and treated D317 GSCs (Figure 4a). Differential expression analysis identified 2247 DEGs in treated D10-0171 cells compared to control cells (1136 upregulated and 1111 downregulated) and 2497 DEGs between control and treated D317 cells (1502 upregulated and 995 downregulated) (Figure 4b). Among the DEGs, 648 were determined to be shared between treated D10-0171 and D317 cells compared to their respective controls, with 363 showing upregulation and 285 showing downregulation (Figure 4b). Volcano plots of the DEGs induced by GZ17-6.02 for D10-0171 and D317 are shown in Figure 4c and Figure 4d, respectively. Among the shared downregulated DEGs are several genes that have been implicated in glioblastoma growth, invasion, and survival, including *EGR1*, *CDH5*, *ABCG2*, *ASCL1*, and *CD109*.This analysis also identified several shared upregulated DEGs with functions that oppose glioblastoma growth and survival, such as *CREBRF*, *BDNF-AS*, and *TLR4*.

We next conducted GSEA using the hallmark gene set collection to identify the pathways downregulated by GZ17-6.02 treatment. Interestingly, GZ17-6.02 downregulated pathways related to cell cycle progression in D10-0171, including E2F targets, G2M checkpoint, and the mitotic spindle (Figure 4e,f). In contrast, GZ17-6.02 downregulated metabolic and inflammatory pathways in D317, such as glycolysis, oxidative phosphorylation, and interferon alpha response (Figure 4g,h).

Data from GSEAs showed a differential effect of GZ17-6.02 on D10-0171 and D317. However, when we analyzed GZ17-6.02’s effect on EGFR signaling using gene set variation analysis (GSVA), we found that GZ17-6.02 significantly downregulated EGFR signaling (Figure 4i) and activation of mitogen-activated protein kinase (MAPK) activity (Figure 4j) in both cell lines.

Phospho-proteomic profiles of control and GZ17-6.02-treated GSCs were obtained with RPPA technology, and Metascape [36] was utilized to analyze the phosphorylated proteins whose expression decreased following GZ17-6.02 treatment. These data are shown in Figure 4k for D10-0171 and Figure 4l for D317. As can be seen, this analysis revealed downregulation of ErbB (EGFR) and mammalian target of rapamycin (mTOR) signaling in addition to several other pathways that promote the growth and survival of glioblastoma in both cell lines following GZ17-6.02 treatment.

We also examined the effect of GZ17-6.02 on recently identified glioblastoma super-enhancer genes *WSCD1*, *EVOL2*, and *KLHDC8A* as well as the enzyme *FADS2*, which mediates *ELOVL2* signaling [8]. These data are shown in Figure 5a,b. As can be seen, GZ17-6.02 significantly downregulates all three super-enhancer genes as well as *FADS2* in both D10-0171 and D317. The effect of GZ17-6.02 on *ELOVL2/FADS2* is noteworthy because blocking the function of *ELOVL2* by inhibiting the enzyme *FADS2* results in downregulation of EGFR signaling, and combinatorial treatment with an EGFR inhibitor and a *FADS2* inhibitor results in a greater decrease in GSC proliferation than either compound alone [8]. Thus, downregulation of both *ELOVL2* and *FADS2* by GZ17-6.02 indicates that GZ17-6.02 inhibits the growth of GSCs by suppressing downstream components of EGFR signaling. We also constructed a GeneMANIA network to provide a visualization of the genes that are co-expressed, co-localized, and closely interact with glioblastoma super-enhancers (Figure 5c). This analysis identified many genes closely associated with the GSC super-enhancers and showed that the top 5 overrepresented functions of these genes were related to fatty acid metabolic and biosynthetic processes as well as monocarboxylic acid, carboxylic acid, and organic acid biosynthetic processes.

Given that the GSC super-enhancer associated gene *ELOVL2* encodes a key enzyme in polyunsaturated fatty-acid synthesis and is crucial in maintaining efficient EGFR signaling in GSCs through its control of fatty-acid elongation [8], we next examined GZ17-6.02’s effect on fatty-acid biosynthesis in D10-0171 and D317. RNA sequencing revealed that GZ17-6.02 treatment resulted in decreased expression of many genes involved in the unsaturated fatty acid biosynthetic process, of which four genes, *PLA2G3*, *ABCD1*, *ANXA1*, and *FADS1*, were found to be downregulated in both GSC lines (Figure 6a, b). We next constructed a GeneMANIA network to highlight the genes that have physical or generic interactions with, or are co-expressed or co-localized with, the shared downregulated unsaturated fatty acid biosynthesis related genes (Figure 6c). Finally, GSVAs were performed to compare the expression of genes involved in unsaturated fatty acid biosynthesis. D10-0171 and D317 GSCs treated with GZ17-6.02 demonstrated a significant decrease in genes related to the positive regulation of unsaturated fatty acid biosynthesis (Figure 6d) and the unsaturated fatty acid biosynthetic processes (Figure 6e).

### 2.4. GZ17-6.02 Inhibits the Growth of D10-0171 GSCs in a Subcutaneous Tumor Model

Given that GZ17-6.02 inhibits the growth of EGFRvIII+ GSCs in vitro and its mechanism of action involves inhibition of several genes involved in glioblastoma growth, the next step was to test whether it is effective in blocking the growth of EGFRvIII+ GSCs in animal models. For these studies, we selected D10-0171 because this tumor is more invasive than D317 and better represents the infiltrative behavior of glioblastoma in humans (Figure 2b). Furthermore, RPPA data indicated increased ErbB signaling in D10-0171 compared to D317 (Figure 3e), and this pathway is known to be disrupted by curcumin and harmine, which are components of GZ17-6.02 [31,32]. The experimental design and tumor growth data are shown in Figure 7. When tumors reached a mean volume of 300 mm^3^, animals were randomized into vehicle control and GZ17-6.02-treated groups. Figure 7b provides tumor size measurements. These data were analyzed using the rate-based tumor/control (T/C) method. This method accounts for random differences in initial tumor volume as it is based on tumor growth rate and allows for the inclusion of animals that were sacrificed early in the final analysis [41]. Using this method, the rate-based T/C was found to be 0.302 (Figure 7b). A rate-based T/C < 0.4 denotes a significant effect of a given compound on tumor growth [41], suggesting that GZ17-6.02 exerts a significant inhibitory effect on tumor growth rate. Furthermore, Kaplan–Meier analysis of tumor development shows that with prophylactic dosing beginning 72 h after tumor inoculation, GZ17-6.02 was effective in delaying tumor formation in mice. The median time to tumor formation was 18 days in untreated mice, and 21 days in GZ17-6.02-treated mice (*p* = 0.021) (Figure 7c).

## 3. Discussion

We demonstrate herein that GZ17-6.02 exerts anti-tumor activity against EGFRvIII+ GSCs. Dose–response and Matrigel invasion assays show inhibition of GCS growth and invasion, while transcriptomic and proteomic analyses provide insight into GZ17-6.02’s mechanism of action. Finally, we show that GZ17-6.02 downregulates several GSC super-enhancer genes, which are key to EGFR signaling and GSC proliferation [8], and demonstrate that GZ17-6.02 suppresses tumor growth in a subcutaneous tumor model.

While our results revealed similar IC_50_ values for GZ17-6.02 in D10-0171 and D317, transcriptomic and proteomic analyses indicate that GZ17-6.02 interacts with these GSC lines differently. In our previous study, we had similarly noted a differential sensitivity of D10-0171 and D317 to growth inhibition by osimertinib [12]. These findings may be attributed to the genomic divergence between D10-0171 and D317 cells. Enrichment analysis of transcriptomic and proteomic data highlights key differences in the baseline molecular characteristics of these cell lines, showing upregulation of cholesterol homeostasis and notch signaling pathways in D10-0171 compared to D317. Functional proteomic analysis comparing the two cell lines demonstrated upregulation of ErbB, insulin, and mTOR signaling in D10-0171. Taken together, these molecular differences likely explain the increased invasiveness observed in D10-0171 compared to D317 (Figure 2b).

Notably, D10-0171 exhibited increased expression of cholesterol synthesis machinery. Alterations in cholesterol metabolism have been gaining increasing recognition as an important mechanism for tumor progression and immune system evasion in various malignancies [42,43]. In glioblastoma specifically, upregulation of cholesterol pathways correlates with poor survival, and depletion of cellular cholesterol results in glioblastoma cell death and decreased tumor growth in mouse models [37,38,39]. Thus, the inclusion of a statin along with GZ17-6.02 could provide even greater growth inhibition of EGFRvIII+ glioblastomas showing molecular characteristics similar to D10-0171. Another differentially upregulated system in D10-017 compared to D317 is the insulin signaling pathway, which plays a role in glioblastoma survival and confers resistance to EGFR inhibitors [44,45]. These differences between D10-0171 and D317 highlight the heterogeneity of EGFRvIII+ glioblastoma and suggest that patients with this genotype should not be treated as a single molecular cohort.

Transcriptomic and proteomic analyses provide additional insight into the mechanisms underlying GZ17-6.02’s anti-tumor activity. Prior studies have shown that GZ17-6.02 interferes with Wnt/β-catenin, Notch, and sonic hedgehog signaling pathways [23]. Additional mechanisms that have been observed include mitigation of invasion, AKT1/EGFR suppression, inhibition of mTOR signaling, and induction of autophagy [14,17]. Consistent with these studies, pathway analysis of RPPA data corroborate GZ17-6.02’s inhibition of mTOR signaling. Prior studies established the role of mTORC1 in promoting invasion and cancer metastases; thus, the robust mTOR inhibition exerted by GZ17-6.02 may underlie its ability to suppress the invasiveness of GSCs [46]. This is further substantiated by gene set enrichment analysis demonstrating downregulation of pathways that are related to cell growth and DNA replication, including E2F targets, G2M checkpoints, and the mitotic spindle in the more invasive D10-0171.

The transcriptomic analysis also revealed that GZ17-6.02 treatment results in suppression of several genes implicated in glioblastoma growth and invasion, such as *EGR1*, *ASCL1*, *CD109*, *ABCG2*, and *CDH5*. *EGR1* codes for a transcription factor involved in healing, fibrosis, and immune responses, that has recently gained recognition for its role in cancer progression [47]. In glioblastoma specifically, *EGR1* is shown to promote the invasion and proliferation of malignant cells, and higher levels of this transcription factor serve as an unfavorable prognostic marker [48]. *ASCL1* is a transcription factor that promotes glioma tumorigenicity through interactions with Wnt and Notch signaling pathways [49]. In murine glioblastoma models, loss of *ASCL1* significantly reduces glioblastoma growth rate and extents animal survival [49]. *CD109* encodes a cell surface protein that directly interacts with glycoprotein 130 to enhance interleukin (IL)-6/signal transducer and activator of transcription 3 (STAT3) signaling to promote tumorigenicity, stemness, and chemoresistance in GSCs [50]. Ablation of *CD109* in glioblastoma models leads to a loss of stemness and a phenotypic shift to a more differentiated astrocyte-like cell [50]. *ABCG2* encodes an ATP-binding cassette transporter that functions as an efflux pump and is associated with chemoresistance in glioblastoma cell lines [51,52]. Additionally, *ABCG2* drives stemness of GSCs by increasing the expression of stem cell markers such as *SOX2* and *MEF* to promote self-renewing capacity [52]. *CDH5* is an endothelial cell marker whose overexpression in malignant gliomas is shown to portray an adverse prognosis, as it promotes the trans-differentiation of GSCs into endothelial-like cells to induce angiogenesis [53].

In addition to suppression of growth-promoting genes, GZ17-6.02 upregulated several genes shown to inhibit the growth and invasion of glioblastoma, including *BDNF-AS, CREBRF,* and *TLR4*. *BDNF-AS* encodes a long non-coding RNA that suppresses the expression of brain-derived neurotrophic factor (BDNF) [54,55]. In glioblastoma, *BDNF-AS* plays a tumor-suppressor role by inducing the degradation of *RAX2* and stabilizing *TP53* transcripts to inhibit proliferation, migration, and invasiveness of malignant cells [56,57]. *CREBRF* also serves as a tumor suppressor gene in glioblastoma as it negatively regulates cyclic-AMP-response element binding 3 (CREB3) to prevent protective autophagy in hypoxic conditions, leading to increased apoptosis [58,59]. *TLR4* is commonly downregulated in glioblastoma [60], as toll-like receptor 4 (TLR4) signaling suppresses retinoblastoma binding protein 5 (RBB5) expression, a protein that acts as a stem cell transcription factor to promote self-renewing capacity. As such, low TLR4 expression by GSCs promotes survival and allows these cells to evade growth suppression by local inflammatory signals [61]. Taken together, the findings from the transcriptomic analysis indicate that GZ17-6.02 has an effect on a wide range of cellular and metabolic pathways, and its ability to inhibit the growth of glioblastoma is likely multifactorial.

Additionally, GSVAs and pathway analysis of protein phosphorylation data demonstrate suppression of EGFR and ErbB signaling. These findings are consistent with curcumin’s and harmine’s known inhibition of EGFR [17]. Of note, interrogation of our transcriptomic data identified reduced expression of the super-enhancer-regulated target gene ELOVL2, which is essential for EGFR signaling. In addition, GZ17-6.02 downregulates the enzyme *FADS2*, which is critical to the elongation of long-chain polyunsaturated fatty acids (LC-PUFA) needed to maintain proper lipid dynamics for EGFR signaling [8]. Thus, GZ17-6.02’s inhibition of ErbB signaling involves both reduced phosphorylation of key mediators and disruption of cell membrane structure. Given that EGFR inhibitors have not yet demonstrated prolonged patient survival [10,11], our results warrant further study of GZ17-6.02 as a potent antitumoral compound to be used in conjunction with EGFR inhibitors.

Limitations of this study include the lack of orthotopic xenograft models to test GZ17-6.02’s ability to inhibit the growth of intracranial glioblastoma, as subcutaneous models may fail to fully recapitulate the tumor microenvironment in the brain [62]. Additionally, no ex vivo analysis was performed on tumor samples from the subcutaneous xenograft model. Future studies should be conducted to test GZ17-6.02’s efficacy in intra-cranial models and to perform further analyses on tumor samples following GZ17-6.02 treatment.

In conclusion, our study shows that GZ17-6.02 inhibits the growth of EGFRvIII+ GSCs both in vitro and in vivo, and that its interactions with the two EGFRvIII+ glioblastomas studied modulate key drivers of multiple signaling pathways. Further preclinical studies are needed to identify the molecular subtypes of EGFRvIII+ glioblastomas that are sensitive to growth inhibition by GZ17-6.02 in both subcutaneous and intracranial models. This information will inform the selection of EGFRvIII+ glioblastoma patients for inclusion in clinical studies designed to evaluate the efficacy of G17-6.02 in this EGFR genotype.

## 4. Materials and Methods

### 4.1. Ethical Statement

All animal protocols used in this study were approved and performed in accordance with guidelines set forth by the Duke University Institutional Animal Care and Use Committee (IACUC) under protocol number A099-20-04. This protocol was approved on 12 August 2020. The reporting in this manuscript follows the recommendations set forth by the ARRIVE guidelines. This study was not preregistered with an official preregistration site.

### 4.2. Cell Culture

EGFRvIII+ GSCs, D10-0171 and D317, were established from patient-derived xenografts as described previously [12]. Cells were maintained in StemPro^™^ media consisting of KnockOut^™^ DMEM/F12 medium (Gibco #12660; Waltham, MA, USA) and supplemented with StemPro^™^ Neural Supplement (Gibco #A1050801; Waltham, MA, USA), 20 ng/mL bFGF (Gibco #0024; Waltham, MA, USA), and 20 ng/mL EGF (Gibco #0314; Waltham, MA, USA), and GlutaMAX^™^-I CTS^™^ (Gibco #A1286001; Waltham, MA, USA). All cells were maintained at 37 °C with 5% CO_2_ and 95% humidity. D317 cells, which form neurospheres, were passaged as follows: cells were centrifuged, washed with phosphate buffered solution (PBS), and incubated in 1 mL Accutase^™^ (Gibco #A1110501; Waltham, MA, USA) for 10 min. Room temperature StemPro^™^ media was added to the cell suspension at a minimum ratio of 1:1 media to Accutase^™^. The cells were triturated by gently pipetting up and down to obtain a single cell suspension, then seeded at a 1.2 × 10^5^ cells/mL density. To passage D10-0171 cells, which grow as monolayers, the cells were washed with PBS and incubated for 5 min with 2 mL of Accutase^™^. At the end of the incubation period, cells were diluted with fresh media and centrifuged. The pellet, consisting of single cells, was resuspended in fresh media to obtain the desired cell concentration. D10-0171 cells were also seeded at a density of 1.2 × 10^5^ cells/mL.

### 4.3. Compounds and Reagents

For in vitro experiments, GZ17-6.02 was prepared as a stock solution in dimethyl sulfoxide (DMSO) and stored at −20 °C. Working solutions were freshly diluted in pre-warmed StemPro media not exceeding a final DMSO concentration of 1%. For in vivo experiments, GZ17-6.02 was taken in Peptamen at a concentration of 30 mg/mL, vortexed, then sonicated for five minutes, and given to mice by oral gavage. All compounds were provided by Genzada Pharmaceuticals (Sterling, KS, USA).

### 4.4. Cell Viability Assays

Compounds were plated using an Echo^®^ 550 Liquid Handler (Labcyte #001-16079; San Jose, CA, USA) onto 384-well plates (Corning #3764; Corning, NY, USA). A Matrix Wellmate Microplate Dispenser (Thermo Scientific # 201-30002; Waltham, MA, USA) was used to dispense cells (1000 cells per well) onto the pre-plated compounds for a final reaction volume of 25 µL with a final DMSO concentration of 1%. The plates were incubated (37 °C, 5% CO_2_, 95% humidity) for 72 h and assayed using CellTiter Glo^®^ reagent (Promega #G7570; Madison, WI, USA). Luminescence values were recorded using a Clariostar Microplate reader (BMG Labtech #0430B0001B; Cary, NC, USA), and the data were analyzed using GraphPad Prism 9.0 Software (San Diego, CA, USA).

### 4.5. Three-Dimensional Invasion Assays

Three-dimensional invasion assays were conducted according to the protocol described by Vinci et al. [63]. Cells were seeded at a density of 10,000 cells per well suspended in 200 µL of StemPro media in ultra-low attachment U-bottom 96-well plates (Corning #CLS3474; Corning, NY, USA). Cells were incubated for 96 h to allow for spheroid formation. Following incubation, 100 µL of media was removed from each well and replaced with thawed Matrigel (Corning #CB40234; Corning, NY, USA). The plates were incubated at 37 °C for 1 h, then 100 µL of media containing 3X final concentration of the desired compound was added to each well. Cells were treated with the indicated compounds for 7 days. All images were acquired using an Olympus CKX53 microscope (Toyko, Japan).

### 4.6. Transcriptomic Analysis

D10-0171 cells were treated with vehicle or GZ17-6.02 for 24 h. Total RNA was isolated using RNeasy plus kit (Qiagen #74034; Hilden, Germany). RNA-seq data was obtained at the Duke University Center for Genomic and Computational Biology and processed using the TrimGalore toolkit (https://www.bioinformatics.babraham.ac.uk/projects/trim_galore (accessed on 4 January 2020)). Only reads 20 nt or longer after trimming were kept for further analysis. Reads were mapped to the GRCh38v93 version of the human genome and transcriptome [64] using the STAR RNA-seq alignment tool [65]. Reads were kept for subsequent analysis if they mapped to a single genomic location. Gene counts were compiled using the HTSeq tool. Only genes that had at least 10 reads in any given library were used in subsequent analysis. Normalization and differential expression were carried out using the DESeq2 [66] Bioconductor [67] package with the R statistical programming environment version 4.1.2. The false discovery rate was calculated to control for multiple hypothesis testing. Gene set enrichment analysis was performed to identify gene ontology terms and pathways associated with altered gene expression for each of the comparisons performed. Gene set variation analysis was conducted with the GSVA R Bioconductor package using the R statistical programming environment version 4.1.2 [68]. Differentially expressed genes were defined as coding genes with a log fold change greater than 1 or less than −1 and a false discovery rate-adjusted *p*-value less than 0.05.

### 4.7. Reverse Phase Protein Array (RPPA) Analysis

Cells were treated with GZ17-6.02 (25 µg/mL) for 24 h, harvested, centrifuged, then washed with ice-cold phosphate buffered saline (PBS). Cell pellets were incubated with lysis buffer (1% Triton X-100, 50 mM HEPES, pH 7.4, 150 mM NaCl, 1.5 mM MgCl_2_, 1 mM EGTA, 100 mM NaF, 10 mM Na pyrophosphate, 1 mM Na_3_VO_4_, 10% glycerol, protease inhibitor (Roche Applied Science #05056489001; Penzberg, Germany) and phosphatase inhibitor (Roche Applied Science #04906837001; Penzberg, Germany)) on ice for 30 min, then clarified by centrifugation at 10,000 r.p.m. After quantifying protein concentration using Bradford assay, lysates were denatured with 4X sodium dodecyl sulfate (SDS) sample buffer (40% glycerol, 8% SDS, 0.25 M Tris-HCl, pH 6.8, 10% (*v*/*v*) 2-mercaptoethanol) and heated at 100°C for 5 min. Samples were stored at −80°C and then sent to MD Anderson’s RPPA core facility (Houston, TX, USA). Gene lists of proteins whose phosphorylation decreased with GZ17-6.02 treatment were analyzed using Metascape [36]. Enrichment analysis was carried out with the following ontology sources: Kyoto Encyclopedia of Genes and Genomes (KEGG) Functional Sets, KEGG Pathways, KEGG structural Complexes, Gene Ontology (GO) Molecular Functions, GO Biological Processes, GO Cellular Components, and Hallmark Gene Sets. Terms with a *p*-value < 0.01, a minimum count of 3, and an enrichment factor > 1.5 were included in the analysis. Network interactions were visualized using Cytoscape.

### 4.8. Subcutaneous Xenograft Tumor Model

To study the effect of GZ17-6.02 in a subcutaneous model, female athymic nu/nu mice aged 8–12 weeks weighing 20–25 g were obtained from the breeding core at Duke University Medical Center. Mice were housed in IACUC compliant Allentown 75 JAG cages with up to 4 mice per cage and had constant access to food and water. Mice were inoculated with 300,000 freshly dissociated D10-0171 cells suspended in 200 µL StemPro media and Matrigel (1:1 *v*/*v*) subcutaneously in the right flank. Once tumors reached a mean volume of 300 mm^3^, mice were randomized into vehicle (*n* = 10) and treatment (*n* = 9) groups. Vehicle control animals received 200 µL Peptamen daily by oral gavage. Mice in the treatment arm received 150 mg/kg GZ17-6.02 daily by oral gavage. In a prophylactic dosing protocol, animals were randomized into control (*n* = 7) and treatment (*n* = 7) groups 72 h after tumor inoculation. Vehicle control animals received 200 µL Peptamen daily by oral gavage. Mice in the treatment arm received 150 mg/kg GZ17-6.02 daily by oral gavage. Animals were observed daily, body mass was measured once weekly, and tumor volumes were measured twice weekly with hand-held Vernier calipers. Tumor volume was calculated with the formula V = (width^2^) × (length)/2. Animals were using carbon dioxide to minimize suffering when they met one of the following humane endpoint criteria: tumor size exceeding 2000 mm^3^, loss of body mass exceeding 20%, or deteriorating body condition. The data were analyzed using a rate-based T/C test [41], and no mice were excluded as this method includes sacrificed animals in the final analysis.

### 4.9. Statistical Analysis

Cell viability data were analyzed using GraphPad Prism 9.0 software (San Diego, CA, USA), and the results were reported as dose–response curves. Comparisons between two groups were analyzed using one-tailed unpaired Student’s *t*-tests.

## Figures and Tables

**Figure 1 ijms-23-04174-f001:**
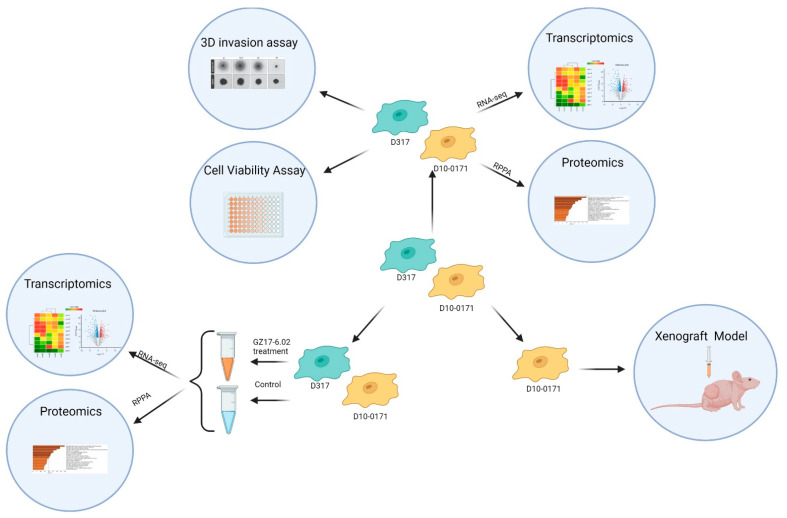
Overall study design. GSC lines D10-0171 and D317 were utilized to test the ability of GZ17-6.02 to inhibit the growth of EGFRvIII+ glioblastoma. The molecular differences between these cell lines were determined with Matrigel invasion and cell viability assays, transcriptomic analysis, and proteomic analysis. These cell lines were also used to characterize the gene and protein expression changes that occur as a result of GZ17-6.02 treatment. The more invasive D10-0171 alone was utilized in a subcutaneous xenograft model. RPPA, reverse phase protein analysis. This figure was created with biorender.com (accessed on 2 February 2022).

**Figure 2 ijms-23-04174-f002:**
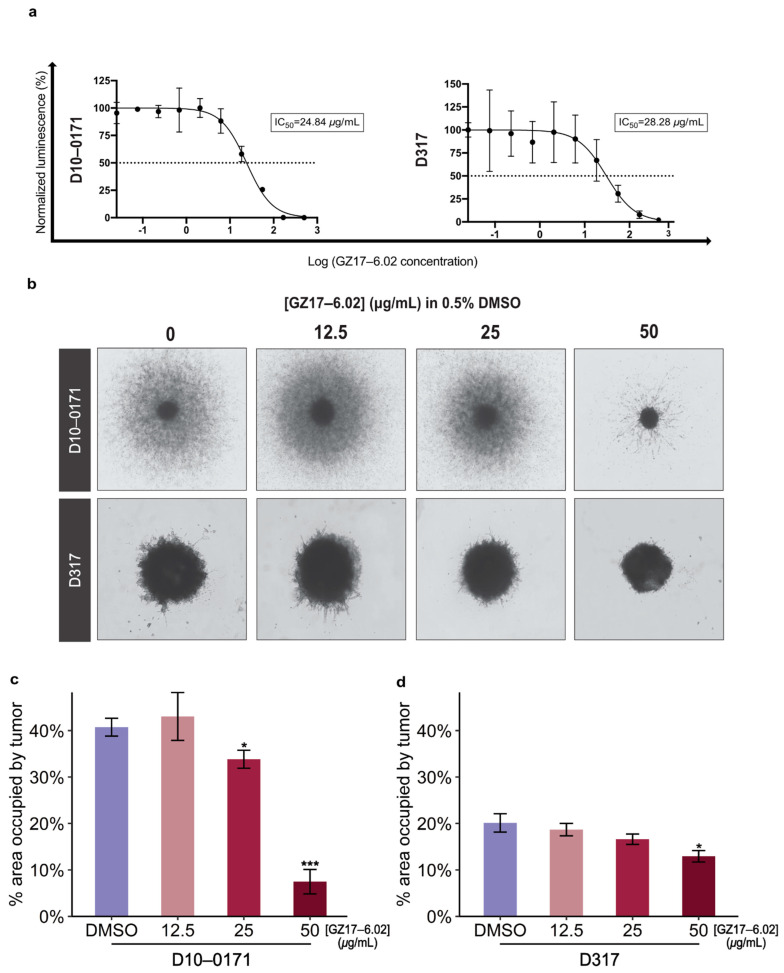
GZ17-6.02 suppresses the growth and invasion of GSCs. (**a**) Representative dose–response curves of two GSC lines tested against various concentrations of GZ17-6.02. (**b**) Representative images of three-dimensional Matrigel invasion assays in a semi-solid matrix showing inhibition of invasive capacity of D10-0171 and D317 by GZ17-6.02. This assay also demonstrates highly invasive behavior of D10-0171 and an apparent lack of invasive capacity of D317. (**c**) Quantification of three-dimensional Matrigel invasion assays for D10-0171 showing percent area occupied by tumor spheroids and their radial projections at varying concentrations of GZ17-6.02. (**d**) Quantification of three-dimensional Matrigel invasion assays for D317 showing percent area occupied by tumor spheroids and their radial projections at varying concentrations of GZ17-6.02. * *p* < 0.05, *** *p* < 0.001. DMSO, dimethyl sulfoxide.

**Figure 3 ijms-23-04174-f003:**
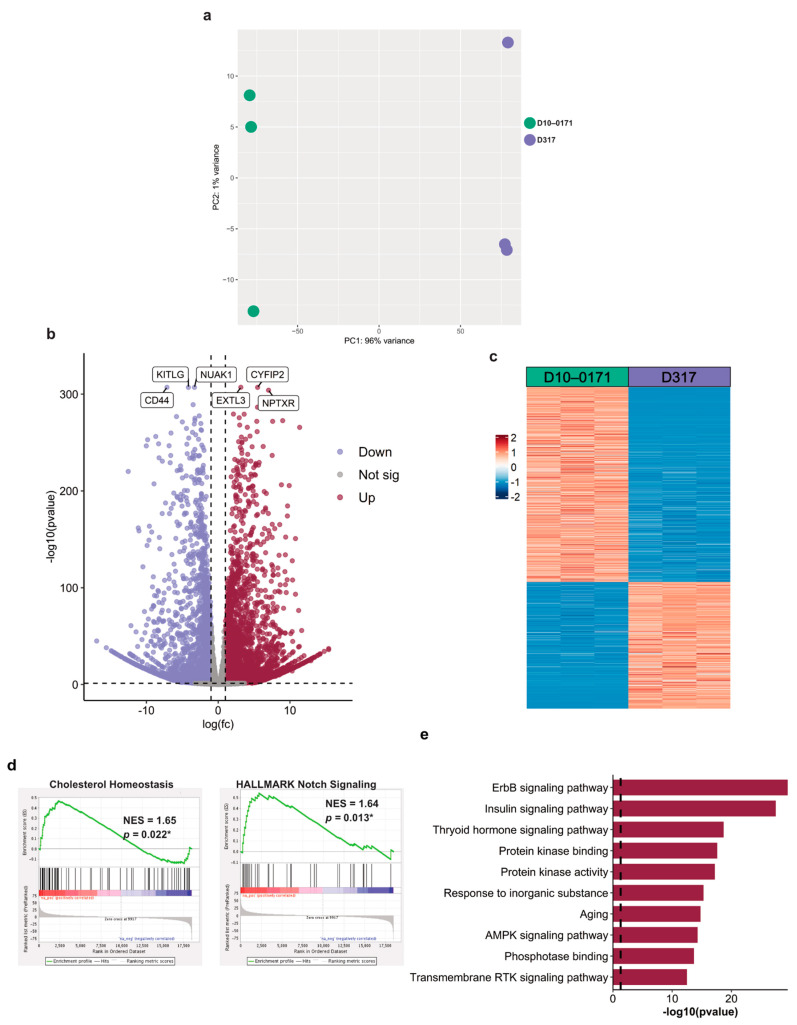
D10-0171 and D317 cells exhibit molecular differences. (**a**) PCA plot of transcriptomic data demonstrates separation of D10-0171 and D317 GSCs. (**b**) Volcano plot of differentially expressed genes between D10-0171 and D317 showing up and downregulated genes in D10-0171 compared to D317. The most significantly up and downregulated genes are labeled. (**c**) Heatmap of differentially expressed genes between the D10-0171 and D317. (**d**) Enrichment plots of significantly upregulated pathways in D10-0171 compared to D317 using hallmark gene set enrichment analysis. *p*-values shown are false discovery rate-adjusted *p*-values. (**e**) Functional enrichment analysis (Metascape [36]) of upregulated pathways across input gene lists of phosphoproteins with increased expression in D10-0171 compared to D317. The dashed line represents a *p*-value of 0.05. * *p* < 0.05, NES, normalized enrichment score; AMPK, AMP-activated protein kinase; RTK, receptor tyrosine kinase.

**Figure 4 ijms-23-04174-f004:**
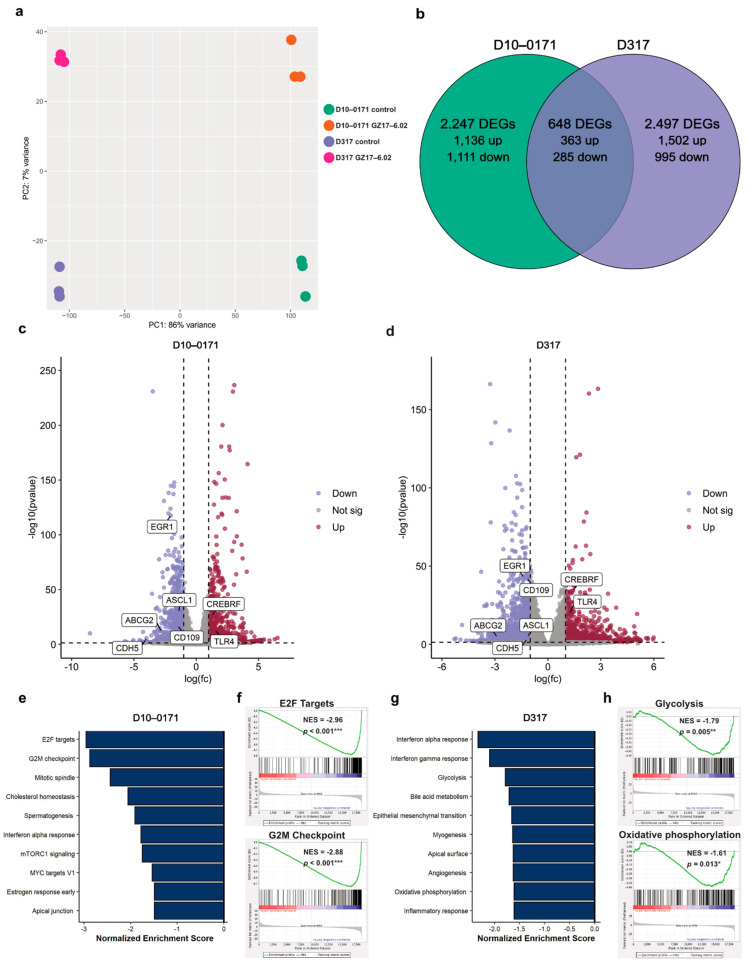
GZ17-6.02 induces changes in gene and protein expression in GSCs. (**a**) PCA plot of transcriptome data demonstrates separation of control and GZ17-6.02 treated D10-0171 and D317 GSCs. (**b**) Venn diagram of shared and unique differentially expressed genes in GZ17-6.02 treated D10-0171 and D317 GSCs. (**c**) Volcano plot of differentially expressed genes in D10-0171 following GZ17-6.02 treatment. (**d**) Volcano plot of differentially expressed genes in D317 following GZ17-6.02 treatment. Shared differentially expressed genes in GZ17-6.02 treated D10-0171 and D317 GSCs that are associated with glioblastoma growth and invasion are labeled on both volcano plots. (**e**) Top 10 downregulated pathways in D10-0171 following GZ17-6.02 treatment identified by GSEA using the Hallmark database as reference. (**f**) Enrichment plots for E2F targets and G2M checkpoint in GZ17-6.02 treated D10-0171 GSCs. (**g**) Top 10 downregulated pathways in D317 following GZ17-6.02 treatment identified by GSEA using the Hallmark database as reference. (**h**) Enrichment plots for glycolysis and oxidative phosphorylation in GZ17-6.02 treated D317 GSCs. (**i**) GSVA comparison of EGFR signaling pathway in control and GZ17-6.02 treated D10-0171 and D317 GSCs. (**j**) GSVA comparison of activation of MAPK activity pathway in control and GZ17-6.02 treated D10-0171 and D317 GSCs. (**k**) Functional enrichment analysis (Metascape [36]) of downregulated pathways across input gene lists of phospho-proteins in D10-171 showing decreased expression following GZ17-6.02 treatment. (**l**) Functional enrichment analysis (Metascape [36]) of downregulated pathways across input gene lists of phospho-proteins in D317 showing decreased expression following GZ17-6.02 treatment. Dashed lines represent a *p*-value of 0.05 * *p* < 0.05, ** *p* < 0.01, *** *p* < 0.001. DEG, differentially expressed gene; NES, normalized enrichment score; mTOR, mammalian target of rapamycin; EGFR, epidermal growth factor receptor; GSVA, gene set variation analysis; TOR, target of rapamycin.

**Figure 5 ijms-23-04174-f005:**
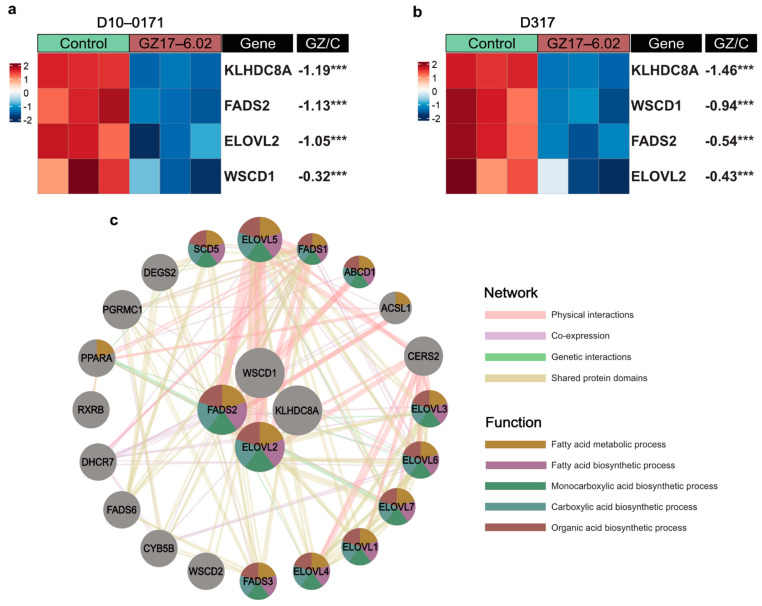
GZ17-6.02 downregulates key GSC super-enhancer genes. (**a**) Heatmap of mRNA expression of GSC super-enhancer genes in control and GZ17-6.02 treated D10-0171 GSCs. (**b**) Heatmap of mRNA expression of GSC super-enhancer genes in control and GZ17-6.02 treated D317 GSCs. The *p*-values shown are false discovery rate-adjusted *p*-values. (**c**) GeneMANIA functional association gene network for GSC super-enhancer genes. Physical interactions and genetic interactions are shown as pink and green lines, respectively, while co-expression is shown as purple lines, and shared protein domains are shown as yellow lines. Stronger associations are shown with thicker lines. Genes are colored based on the top 5 over-represented shared function. Genes colored yellow share functions related to the fatty acid metabolic process, genes colored purple share functions related to the fatty acid biosynthetic process, genes colored green share functions related to the monocarboxylic acid biosynthetic process, genes colored blue share functions related to the carboxylic acid biosynthetic process, and genes colored red share functions related to the organic acid biosynthetic process. Fold change shown is log base 2-fold change. *** *p* < 0.001. GZ, GZ17-6.02; C, control.

**Figure 6 ijms-23-04174-f006:**
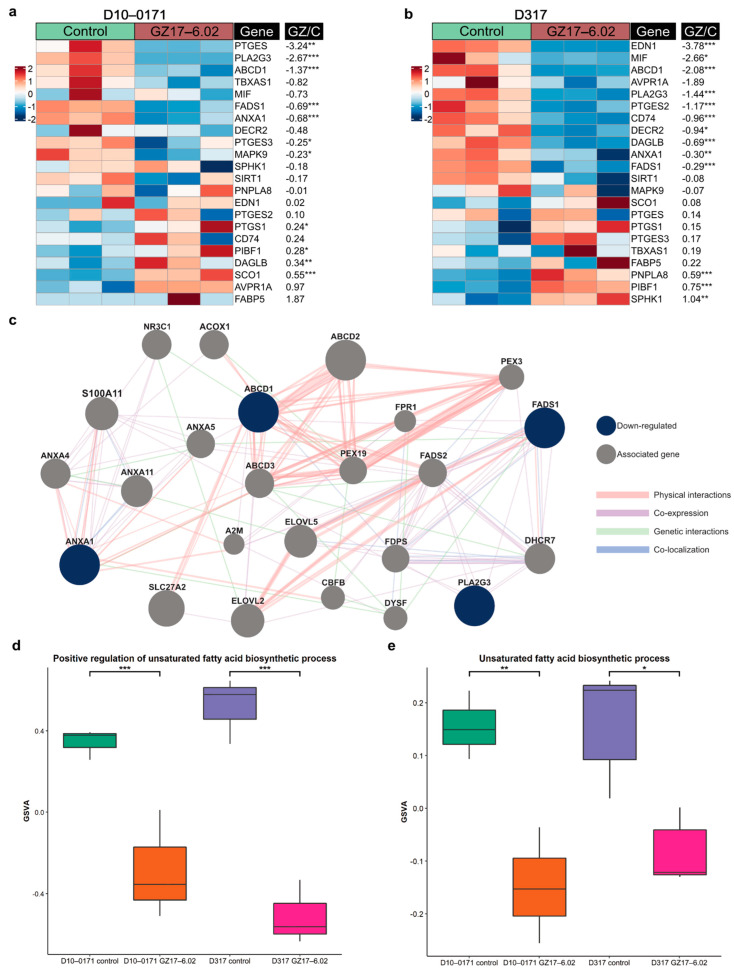
GZ17-6.02 downregulates unsaturated fatty acid biosynthesis. (**a**) Heatmap of mRNA expression of select unsaturated fatty acid biosynthesis related genes in control and GZ17-6.02 treated D10-0171 GSCs. (**b**) Heatmap of mRNA expression of select unsaturated fatty acid biosynthesis related genes in control and GZ17-6.02 treated D317 GSCs. The *p*-values shown are false discovery rate-adjusted *p*-values. (**c**) GeneMANIA functional association gene network for select unsaturated fatty acid biosynthesis related genes. Genes shown in blue are downregulated by GZ17-6.02 in both D10-0171 and D317, while associated genes are shown in gray. Physical and genetic interactions are shown in pink and green, respectively, while co-expression and co-localization between genes are shown in purple and blue, respectively. Thicker lines indicate stronger associations. (**d**) GSVA comparison of positive regulation of unsaturated fatty acid biosynthetic process in control and GZ17-6.02 treated D10-0171 and D317 GSCs showing a significant decrease in both following GZ17-6.02 treatment. (**e**) GSVA comparison of unsaturated fatty acid biosynthetic process in control and GZ17-6.02 treated D10-0171 and D317 GSCs showing a significant decrease in both following GZ17-6.02 treatment. Fold changes shown are log base 2-fold changes. * *p* < 0.05, ** *p* < 0.01, *** *p* < 0.001. GZ, GZ17-6.02; GSVA, gene set variation analysis; C, control.

**Figure 7 ijms-23-04174-f007:**
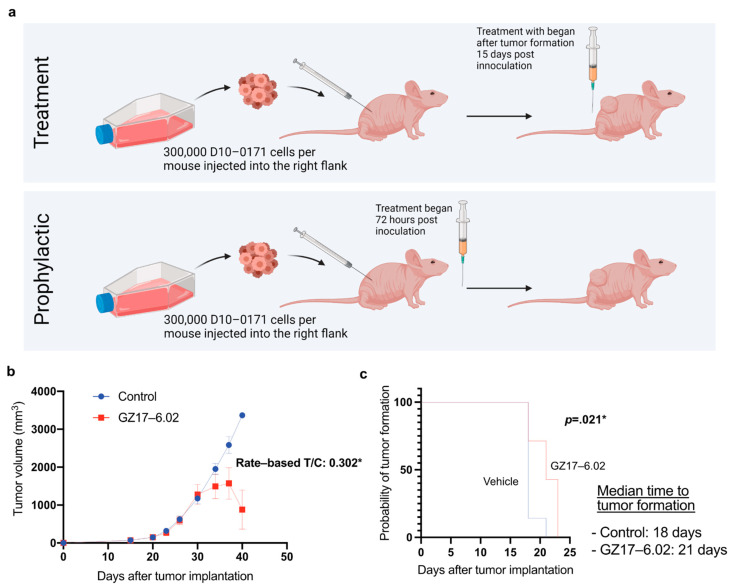
GZ17-6.02 demonstrated anti-tumor activity in a subcutaneous tumor model. (**a**) Flow chart depicting the study protocol with athymic NU/NU mice. This figure was created with biorender.com (accessed on August 20, 2021) (**b**) Average tumor volume per mouse plotted over time in control and treated mice, and analysis of tumor volume data using the rate-based T/C method. A rate-based T/C value below 0.4 indicates that the compound had a significant effect on tumor growth. (**c**) Kaplan–Meier analysis of tumor onset demonstrating that GZ17-6.02 delays tumor formation. * denotes a significant value. n, number of animals; T/C, tumor/control.

## Data Availability

The data that support the findings of this study are available from the corresponding author M.M.K., upon reasonable request.

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
