# Peer review of "GZ17-6.02 Inhibits the Growth of EGFRvIII+ Glioblastoma"

_ijms, 2022, doi:10.3390/ijms23084174_

Round 1

Reviewer 1 Report

The authors of this work evaluated the action of a molecularly targeted drug against EGFRVIII. Firstly, the authors calculated the the IC50 and tested the ability to reduce invasion on two glioblastoma stem cell lines, verifying a different response and greater invasiveness of the D10-0171 cell line. A first transcriptomic analysis highlighted different gene and molecular expression profiles. Subsequently, proteomic and transcriptomic analyzes revealed that treatment with GZ17-6.02 induces different effects on the cell cycle, metabolism and pathways of EGFR; a downregulation of key GSC super-enhancers and unsaturated fatty acid biosynthesis genes was also reported. Finally, GZ17-6.02 was shown to inhibit tumor growth in a mouse heterotopic model.

  1. Section “1. GZ17-6.02 inhibits the growth and invasion of EGFRvIII+ GSCs” includes transcriptomic data without GZ17-6.02 treatments. Therefore, I suggest separating the section about growth and invasion assay from trascriptomic analysis of the two cell lines.
  2. Figures about genomics and proteomics analysis are not well explained and described. A lot of information is implicit and therefore data are unclear.
  3. Figure 6b does not seem appropriate since macroscopically the tumor masses cannot show significant differences.
  4. The study seems incomplete, only a single in vitro test was done to declare the different invasive pattern of the cell lines; moreover, this test does not show any quantitative data but only representative images.
  5. The major lack of the study is represented by the final part of the work. After the creation of heterotopic models, the conclusion is related only to the reduction of the tumor mass after treatment. No ex vivo evaluation was performed and the authors did not detect any targets or biomarkers related to those listed in the genomic and proteomic expression profiles.
  6. I suggest improving the resolution and quality of the figures.

Reviewer 2 Report

In this paper, Choi and colleagues investigated the effect of GZ17-6.02, a novel anti-cancer agent, on two EGFRvIII+ glioblastoma (GBM) stem cells, D10-0171 and D317, both in vitro and in vivo, in a subcutaneous xenograft model of GBM.

Overall the manuscript is well written; the introduction provides sufficient background along with relevant literature.

A major comment concerns the in vivo model used in the study, which is a subcutaneous xenograft tumor model. However, while subcutaneous xenografts are less technically challenging than orthotopic xenografts and are easily passaged in vivo, orthotopic xenografts more closely mimic the clinical situation of the disease. Although the response to drugs can be assessed easily in subcutaneous tumor xenografts, however in this GBM model some key aspects of the disease, such as the prominent tissue invasion, are absent (Hofmann, 2015). In fact, GBM rarely grows outside of the brain in humans, which further underlines the importance of the brain microenvironment in studying key aspects of this disease (Irtenkauf et al., 2017). Thus, GBM pathology would be better reproduced in the mouse brain than in an ectopic environment. Maybe the Authors should discuss this aspect, which represent a limit of the study.

In the Material and Methods section, it is not clear whether curcumin, harmine, and isovanillin are some natural components of the GZ17-6.02 anticancer agent (line 380); some more details regarding this novel anticancer agent are needed, even in the introduction (line 56 or 64).

Why “curcumin, harmine, and isovanillin were prepared as stock solutions in dimethyl sulfoxide (DMSO)” (lines 380-381)? In the paper, the Authors did not treat the cells with either curcumin or harmine or isovanillin alone.

Round 2

Reviewer 1 Report

The authors have sufficiently addressed the critical issues of the work, certainly improving the quality of the manuscript.

Some minor suggestions:

  1. The graphs in figure 2c should be shown separately if there is no statistical comparisons between the invasion of two cell lines
  2. I suggest to remove figure 3a and its description in section 2.2 since it shows the same data illustrated in Figure 2c with vehicle.
  3. A minor suggestion would be to use the acronym GBM for “glioblastoma”. Furthermore, the acronym GSCs has been used for "glioblastoma stem cells" but in some parts it is written in full

Reviewer 2 Report

The Authors have addressed all the concerns raised by the Reviewer. I have no more academic questions.

Author Response

Thank you for your comments and for taking the time to review our manuscript.